# Alternative Treatments for Emotional Experiencing and Processing in People with Migraine or Tension-Type Headache: A Scoping Review

**DOI:** 10.3390/healthcare12131302

**Published:** 2024-06-29

**Authors:** Alessandra Marelli, Licia Grazzi, Marina Angela Visco, Pietro Crescenzo, Alessandra Bavagnoli, Cristal Sirotich, Venusia Covelli

**Affiliations:** 1Faculty of Psychology, e-Campus University, 22060 Novedrate, Italy; marina.visco@uniecampus.it (M.A.V.); alessandra.bavagnoli@uniecampus.it (A.B.); cristalsirotich@gmail.com (C.S.); venusia.covelli@uniecampus.it (V.C.); 2Neuroalgology Unit, Headache Center, Fondazione IRCCS Istituto Neurologico Carlo Besta, 20133 Milan, Italy; licia.grazzi@istituto-besta.it; 3Department of Education, Psychology and Communication Sciences, University of Bari ‘Aldo Moro’, 70121 Bari, Italy; pietro.crescenzo@uniba.it

**Keywords:** headache, migraine, non-pharmacology treatment, complementary and alternative medicine, emotion management, emotional disclosure, emotional expression, emotional control

## Abstract

This narrative review aims to summarize the use of alternative treatments (e.g., relaxation training, meditation, written intervention) for emotional expression, processing, control, or management in patients with migraine and tension-type headaches, which the previous literature has shown to be related to migraine pain perception and headache symptoms. Online databases including PubMed, Scopus, Web of Science, PsycINFO, and Medline were searched to identify studies published between 2000 and 2023. A descriptive synthesis of the included studies was conducted. We included seven articles after screening 1.173 records. A total of 610 patients with a diagnosis of migraine or tension-type headache, and an average age of 19–45.5 years (68–90.4% females) were recruited in the selected studies. Overall, the results show that alternative approaches to headache treatment contribute to the management, reduction, or control of negative emotions and at the same time have a positive impact on pain perception and headache symptoms. However, in some cases, the effects are more promising than others, depending on the peculiarities or limitations of each approach considered. This review provides useful insights from a methodological point of view for future studies on the management or control of negative emotions in patients with migraine and tension-type headache.

## 1. Introduction

Migraine and tension-type headaches (TTHs) are considered the most common types of primary headache disorders [1], along with significant work- and family-related disability [2]. Worldwide, 46% of the adult population have an active general headache disorder, 11% suffer from migraine, 42% from tension-type headache, and 3% from chronic daily headache, as reported by the International Headache Society [1,3]. Several studies have described the impact of primary headaches on people’s health and quality of life. It negatively affects people’s social functioning and relationships, reduces their quality of life, and interferes with the performance of normal daily activities [4,5,6].

Alongside pharmacological treatments, research over time has also focused on non-pharmacological interventions for the treatment of headaches and the reduction in their associated symptoms, which are causes of severe discomfort in patients [7]. Non-pharmacological interventions have arisen precisely for the management of those patients for whom oral pharmacological treatments were poorly tolerated or resulted in chronic headaches and/or drug abuse. These treatments range from commonly used and easily accessible nutraceuticals, such as magnesium and riboflavin, to the new area of neuromodulation interventions and biofeedback techniques [8], or mindfulness-based relaxation and meditation interventions [9,10,11]. In addition to this therapy, established psychological therapies, such as cognitive behavioral therapy (CBT) [12], Acceptance and Commitment Therapy (ACT) [13], and psychotherapy intervention [14], are also proving to be effective therapies, with a reduction in pain and a decrease in the frequency of migraine headaches.

The spread of non-pharmacological treatments also depends on an increased focus on psychological, psychopathological, and psychiatric disorders. People with frequent headaches appear to be more physically reactive to stress, with increased depression and anxiety symptoms, which play an important role in pain perception and headache duration [15,16,17]. Moreover, people with TTH, in addition to reaching high depression scores, may have difficulty expressing emotions [18]. With this in mind, several studies support the theory that people with frequent migraines are significantly associated with increased negative emotions and negative emotional states (such as irritability, worry, anger, hostility, and fear), which in turn play an important role in the emotional experience of pain [19,20,21]. As negative emotions can trigger and prolong migraines, we considered how the growing literature attests to the importance of emotional processes in the management of chronic pain. Accessing, experiencing, and expressing negative feelings is potentially adaptive: it may enable habituation, provide valuable information about behavioral tendencies, allow for the assimilation of cognitions and emotions, and ultimately lead to the resolution of the stressor [22,23,24]. The management, reduction, or control of negative emotions has a positive impact on pain perception and headache frequency. This has led to increased attention to non-pharmacological interventions on emotion management that can reduce the frequency and severity of primary headache [25], as well as in other chronic pain conditions, such as fibromyalgia and rheumatoid arthritis [26,27].

Given the growing attention to alternative treatments alongside pharmacological ones, and the increasing focus on the impact of emotional effects on reported pain and physical symptoms in people with primary headache, we decided to conduct a narrative review to summarize which alternative approaches had been used for the management and processing of emotions in patients with primary headache and what their effects are. We also provide methodological details about our study design and procedure, and insights into this study’s limitations that future studies could consider to fill the gaps in the literature.

## 2. Materials and Methods

### 2.1. Search Strategy

The search and selection of studies related to our topic was carried out in accordance with the PRISMA guidelines (Preferred Reporting Items for Systematic Reviews and Meta-Analyses) [28]. Five electronic databases (PubMed, Scopus, PsychInfo, Web of Science, and Medline) were searched for studies published between 1 January 2000 and 31 May 2023. Search strategies were customized for each database by combining the term headache (migraine OR tension-type headache OR tension headaches) with the following keywords related to non-pharmacological treatment (non-pharmacology treatment OR complementary alternative medicine OR behavioral treatment) and related to emotions (emotion* OR emotional expression OR emotional disclosure OR emotional management OR emotional processing OR emotional control). The search was carried out in the titles and abstracts of the studies. For non-pharmacological interventions, we do not consider psychotherapeutic treatments for which special attention would be needed in future reviews.

### 2.2. Study Inclusion and Exclusion Criteria

Studies were included if they met the following criteria: journal articles in English with an abstract and full text available, research articles focused on emotional management/expression in combination with non-pharmacological treatments in patients with migraine or tension-type headache (with the exclusion of psychotherapy interventions); populations with a diagnosis of chronic primary headache (migraine, tension-type headache, or mixed headache), and peer-reviewed research articles. To retrieve information on the review topic, qualitative studies were included.

The following records were excluded: books or book chapters, review papers, case reports/case series, study protocols, commentaries, letters, editorials, conference abstracts or posters, psychometric studies (developments or validations of questionnaires or scales), and papers not focused on the review topic.

### 2.3. Study Selection and Data Extraction

All abstracts identified by electronic searches were independently scrutinized by two researchers (AM and VC), and the full texts of relevant studies were retrieved. Where information in the abstract was not sufficient to classify a study, the full text was read. To ensure the quality and consistency of data extraction, 20% of the abstracts were randomly selected for a second check by another two reviewers (AB and CS), who were blinded to the decisions of the first two. Each reviewer had to rate the study as excluded, eligible, or ambiguous. The full texts of studies that were judged eligible or ambiguous were then analyzed, and 10% of the full texts were double-checked by two reviewers independently.

## 3. Results

A total of 1.173 records were initially retrieved. After duplicates had been eliminated (N = 226), a total of 947 records were screened, 20 full texts were read, and seven studies were selected, as shown in Figure 1. Studies were excluded if they were not written in English or not available, and if they did not deal with the topic of the present review. The main characteristics of the included studies are reported in Table 1.

### 3.1. Participants

A total of 610 patients with a diagnosis of migraine or tension-type headache were recruited in the selected studies. In each study, participants were predominantly females, ranging on average between 68% and 90%. All studies involved participants with primary headache, selected with different criteria. In two studies [29,32], at least two migraine episodes occurred in the previous month. One study [31] considered at least ten episodes per month; one study [34] considered at least one episode per month; the remainder did not specify. Finally, another study [30] involved patients with headaches without specifying the type, and two studies [32,33] involved people with migraine as well as tension-type headache. Apart from [31], which involved the general population, and study [35], which involved patients recruited from a hospital, the remaining studies involved college students.

### 3.2. Design and Alternative Treatment Description

The included studies included a longitudinal randomized trial [29], four randomized trial studies [30,32,33,34], a clinical intervention study [31], and a quasi-experimental study [35]. Six studies were conducted in the USA, and one was conducted in Iran [35].

In terms of alternative treatments used in the experimental conditions, five studies considered relaxation training (RT) as a comparison treatment combined with other alternative approaches [29,30,32,33,34]. Two studies considered the written emotional disclosure (WED) intervention in combination with RT [33,34], two studies included different forms of meditation interventions (MIs) in combination with RT [29,32], and one study [30] included anger awareness and expression training (AAET) in combination with RT. Two studies considered only one alternative treatment: one study [31] evaluated a meditation intervention (Buddhist Loving Kindness meditation practice), and another study [35] evaluated compassion-focused therapy (CFT). More precisely, RT consist of a technique using a variety of strategies, such as progressive muscle relaxation, deep or controlled breathing, guided imagery, distraction, and sometimes biofeedback, aimed at reducing negative emotions and physiological arousal, and increasing calmness, to improve the subject’s health [10]. In WED treatment, the participant is asked to write about facts and their feelings about a stressful experience and to try to write about the same topic repeatedly, creating a narrative and relating the stressor to their life. This technique considers the access, experience, and expression of negative emotions as potentially adaptive, because they allow for the assimilation of cognitions and emotions and lead to the resolution of the stressor [36]. Meditation has also received considerable attention to help patients with chronic pain regulate their emotional reactivity to pain [37]. Some meditation practices use a meaningful phrase, a mantra, or a cognitive focus to direct the mental exercise. Others focus on the benefits of repeating “any calming, melodious sound”, no matter its meaning to the individual [29]. Different forms of meditation include spiritual, secular, and Buddhist Loving Kindness meditation [37,38]. The anger awareness and expression training (AAET) intervention was developed by Slavin-Spenny and colleagues [30]. It consists of a brief group-based intervention that seeks to reduce stress by helping people become aware of and accept their anger as normal and adaptive. Moreover, it helps people to experience it subjectively and bodily, and to use the anger to motivate adaptive behavior, and particularly assertive communication in stressful relationships. The intervention is brief (three sessions) and held in groups, so as to reduce costs and lead to a higher uptake or adherence among patients. Finally, CFT consists of a protocol composed of multiple sessions that take the participant to a deep awareness of the suffering of the self and others, stimulating the desire and attempt to heal this suffering. The person is stimulated to focus on the presence of the pain and its causes; instead, healing focuses on the will and actions to try to reduce the suffering [35].

The specificity of each treatment procedure considered in the selected studies will be described in the next section.

### 3.3. Procedure and Main Results

The first longitudinal randomized controlled trial study conducted by Wachholtz and colleagues [29] aimed to compare the effects of three active management forms of meditation vs. a cognitive distraction meditation on headache-related pain and emotions. Participants were randomly assigned to four different groups. Three groups were designated as active management groups (which taught participants an active emotional or physiological approach to managing migraine), and one group was designated the cognitive distraction group (which was provided with a cognitive activity). Participants in each group were asked to practice the proposed technique for 20 min a day for 30 days in a quiet, un-distracted environment. They were encouraged to record their practice in a diary, which they were asked to email to the researcher every day. In the diaries, participants had to report how bad the headache had been during the previous 24 h and how they felt (happy, relaxed, sad, or angry) using a seven-point Likert scale. The authors measured the effect of treatment by participants’ self-reported pain levels and experienced emotions (anger and happiness) over the 30 days of the diary. The active groups showed significant reductions in anger and migraine pain over time: the three active management groups using a meaningful cognitive focus that included body-focused, self-esteem-focused, or spiritually focused forms of meditation reported a significant decrease in anger and pain over the course of 30 days, with the most significant change occurring after 20 days of practice; whereas the cognitive distraction group (provided with a cognitive activity, and not meaningfully focused) was less effective in reducing patients’ reported pain or improving patients’ negative mood.

The randomized trial conducted by Slavin-Spenny and colleagues [30] aimed to compare the effects of a brief group-based anger awareness and expression training session to relaxation training on headache management and negative emotions. Having completed baseline measures, participants were assigned to three different groups: two experimental conditions and a control group. Each of the two experimental conditions consisted of three intervention sessions. The AAET experimental condition consisted of (1) experiential exercises in recognizing, experiencing, and expressing anger, (2) learning to communicate anger adaptively, and (3) role-playing exercises to practice self-assertive communication. The experimental condition for RT consisted of (1) teaching participants progressive muscle relaxation; (2) teaching participants deep breathing relaxation and short applied relaxation exercises; and (3) teaching guided imagery relaxation and exploring how to incorporate relaxation into daily routines. Emotional processing and expression were assessed using the four-item emotional approaches scale [39]. The author measured the effects on headache frequency; severity and duration; and headache disability and psychological symptoms. Overall, both interventions (AAET and RT) were at least effective in improving headache outcomes compared to the control group. Regarding the manipulation of affect control reactions, the RT condition reduced negative affect more than the AAET condition, although negative affect decreased significantly in both conditions. On process measures, AAET reduced alexithymia and increased emotional processing and assertiveness, and led to greater assertiveness and emotional processing than the control.

In their clinical intervention-based study (single-group repeated-measures pre/post design), Tonelli and Wachholtz (2024) [31] aimed to explore the efficacy of the Buddhist Loving Kindness meditation practice in reducing both migraine-headache symptoms and emotional tension (negative affective state). A sample of 27 migraineurs attended a 20 min group-based guided meditation session and completed a pre- and post-intervention survey to assess migraine pain and emotional tension. The Numeric Rating Scale (NRS-11) [40] in its original form was used for the assessment of pain, and it was altered (not specified) to assess emotional tension. These assessments were considered as outcome measures, including pain and emotional tension. Correlation analyses showed a strong association between levels of pain and emotional tension. After 20 min of Buddhist Loving Kindness meditation practice, there was a significant decrease from pre- to post-treatment in both reported pain and emotional tension. To be precise, there was a 33% decrease in pain and a 43% decrease in emotional tension from pre- to post-meditation data.

The randomized trial conducted by Wachholtz and Pargament [32] aimed to compare spiritual meditation (SM) to internal secular meditation (ISM), external secular meditation (ESM) and relaxation training (RT) in enhancing pain tolerance and reducing migraine–headache-related symptoms. Participants were randomly divided into four groups (SM, ISM, ESM, and RT) and completed in 20 min sessions a day for one month for the condition assigned. The Positive and Negative Affect Scale (PANAS) [41] was used to assess positive and negative affect. The authors measured the effect of treatment using pain measures (pain tolerance and headache frequency and severity) collected by participants recorded in their practice diaries and some psychological measures. The results showed that participants who practiced 20 min of meditation every day for one month in all forms of meditation considered (SM, ISM, and ESM) reported significant decreases in pain tolerance, headache-related self-efficacy, daily spiritual experience, and existential well-being. For negative affect, the SM group reported a significantly greater drop in negative affect over the course of the study than the ISM or ESM groups. Concerning positive affect, three of the four groups (SM, ISM, and RT) experienced some modest improvement in their positive affect, but this was not significant.

The randomized trial conducted by D’Souza and colleagues [33] aimed to compare the effects of RT and WED with respect to mood, headache frequency, severity, disability, and physical symptoms. Participants completed baseline health measures and were then assigned to one of three intervention groups: two experimental groups (WED and RT) and a writing control group. Each experimental condition consisted of four sessions of 20 min each, delivered over two consecutive weeks. Immediately before and after each intervention session, participants rated their current mood using an abbreviated version of Positive and Negative Affect Schedule-X (PANAS-X) [42]. At the 1- and 3-month follow-ups, participants returned to the laboratory to complete health measurements. The authors considered as primary outcomes headache frequency and severity, as participants reported in their daily diary, and headache disability measured by the Migraine Disability Assessment Scale (MIDAS) [43]. The results showed that RT produced immediate increases in calmness, whereas WED led to an immediate increase in negative mood, in both samples (TTH and migraine). Statistical analyses reported that in the TTH sample, RT led to an improvement in headache frequency and headache disability, compared with the other two groups (WED and control). WED did not affect the tension-type headache sample. Regarding migraine, RT led to an improvement in pain severity compared with the control group, whereas WED had no effect.

The randomized trial conducted by Kraft et al., 2008 [34], aimed to compare WED to RT vs. control writing, and to test the moderate effects of emotional skills and headache management self-efficacy. Participants were asked to complete baseline moderator and health measures. They were then randomized into two experimental groups, in which participants received WED and RT, and a control group. In each group, participants completed four 20 min laboratory sessions over two weekly sessions. After 1 and 3 months, follow-up assessments were scheduled. The eight-item emotional approach coping scale (EAC) [39] was used to gather emotional measures. The results showed a high level of approach to emotional coping predicted improvement after WED compared with RT and the control group. Compared with the control group, a low level of self-efficacy for headache management was a strong predictor of improvement after both interventions. Analyses of available three-month results revealed more robust effects on headache management self-efficacy, but slightly more attenuated effects on emotional management.

Finally, the quasi-experimental study conducted by Barchakh et al. [35] aimed to investigate the effectiveness of CFT for improving emotional control and reducing the severity of pain in patients with migraines. Participants were randomly divided into experimental and control groups. Patients in the experimental group received eight 90 min sessions of CFT. Both experimental and control groups completed the post-test after the training sessions. The Emotional Control Questionnaire [44] was used for the assessment of emotional inhibition, aggression control, rumination, and positive control. The emotional control and pain level were measures, respectively, within the Emotional Control Questionnaire (ECQ) [45] and Questionnaire of von Korff for grading the severity of chronic pain [46]. The authors considered as primary outcomes headache frequency and severity, as participants reported in their daily diary, and headache disability, measured using the Migraine Disability Assessment Scale (MIDAS) [43]. Negative and positive affect for the last month was rated on the Positive and Negative Affect Schedule [41]. The results reported a significant effect of CFT on improving emotional control and its subscales (emotional inhibition, aggression control, rehearsal, and benign control), as well as decreasing pain severity in patients with migraine. The study results supported the main hypothesis of the study concerning the effectiveness of CFT for improving emotional control.

### 3.4. Limitations

A summary of the main limitations of the studies included for this review are presented in Table 2. Consideration of the limitations is important for allowing future research to be directed. Whenever possible, larger samples of patients with primary headache, selected by screening, and avoiding self-report measures for diagnosis, should be enrolled, so that differences in primary headache type can be detected over the long term. Also, we recommend the provision of control groups and longer follow-up periods, so that treatment efficacy can be better assessed. Future studies could capture these limitations to fill the voids in the literature.

## 4. Discussion

The results of the studies analyzed show that the alternative approaches considered (RT, WED, different forms of mediation, written emotional disclosure, AAET, and CFT) might help to manage, reduce, or control negative emotions. However, these promising effects depend on the specificity or limitations of each approach considered.

The alternative treatments analyzed might be divided again into those that invite patients to partake in relaxation, internally focused and external focused (SM) meditation, or self-compassion (CFT), and those such as AAET and WED that work on awareness, expression, and the processing of negative emotions. This distinction is in line with what Lumley et al. [22] summarized: when it comes to psychological interventions, there are two approaches in the literature regarding emotional experiencing and processing. The first approach includes the treatments trying to downregulate, minimize, or avoid emotions (e.g., relaxation or breathing retraining), and the second includes the treatments trying to facilitate the awareness, expression, and processing of negative emotions (e.g., virtual reality exposure or experiential techniques).

We proceed to discuss the main effects of each alternative treatment in relation to previous studies; then, we discuss the comparison of various treatments.

For specific treatments, three studies [29,31,32] focused on the use of different forms of meditation in reducing negative emotions in patients with primary headaches, with a consequent reduction in pain perception and headache frequency. These results are largely consistent with previous research in the literature, showing that using and repeating meaningful sentences during meditation practice can lead to a greater sense of comfort (in terms of mental relaxation and reduction in negative emotions) in migraineurs [47,48,49]. Moreover, a specific form of meditation (spiritual meditation) enhances the ability to feel able to control one’s pain (self-efficacy) through meditative intervention and to reduce pain intensity by being able to carry out one’s daily tasks. This correlates with a lower level of negative emotions caused by pain and a more positive mood, in line previous studies [50,51]. One study [35] is part of the line of research that has investigated alternative approaches, such as CFT, that have a significant impact on the improvement in emotional control and the reduction in pain severity in patients with primary headaches. The results appear to be consistent with previous studies that demonstrated the effectiveness of CFT in improving emotional control in migraine patients [52,53,54,55,56]. People with emotional control can identify their emotions, understand them, and express their emotional states to others more effectively. Instead, AAET and WED work differently (on awareness, expression, and the processing of negative emotions). AAET acts by activating emotions and increasing arousal, assertiveness, and emotional processing, and reducing alexithymia (a construct that has both trait and state components and that has been shown to decrease in response to emotion-focused interventions). Two studies [33,34] considered WED, which was found to be ineffective for TTH and migraine patients. The benefits of WED intervention may depend on the type of sample and the methodology used: having emotional competence and low self-efficacy are predictors of WED ineffectiveness, characteristics that were not considered in the sample studied [34,57]. On the other hand, results from another study have shown how alexithymia (a lack of emotional understanding, expression, and introspection) predicts poorer responses to WED [58]. Moreover, WED and other similar techniques have a more specific applicability, because the people who might benefit from them must not only have an unresolved stressful event to recount, but also the motivation to recount it repeatedly, the ability to tolerate the temporary negative mood induced by the recounting, and the ability to translate their experience into emotional words by engaging in appropriate cognitive reworking [36]. Consistent with this are the results of [34], which showed that the individual possession of emotional skills enhanced the benefits of WED. People with low narrative motivation or have a limited ability to express and reframe their emotions may find WED unattractive or have difficulty identifying stressors, disclosing their feelings, and generating cognitive or affective change [59,60].

In the studies we analyzed, when the comparison was between treatments “deactivating” emotions, greater positive effects on headache symptoms, pain perception, and emotional tension were reported (for example, the greater impact of spiritual meditation compared with other forms of meditation or RT). On the other hand, when AAET was compared to RT, or WED was compared to RT, the results showed greater effects of RT in decreased arousal and reduced negative mood, whereas for AAET and WED, an increase in emotional activation and negative mood was found. At the same time, an increase in emotional processing and expression, at least as far as AAET is concerned, was observed. In WED, this effect has not been investigated. We know that it increases emotional activation and negative mood, but we do not know whether it may have an effect on emotional expression or processing or assertiveness, because it has not been investigated. Indeed, in [34], high measures on the emotional approach coping scale may predict greater effects of WED on headache frequency, pain severity, functional and emotional disability, and negative–positive affect.

Thus, while action could be taken to foster emotional skills, better investigation could be carried out for which patients benefit more from emotional expression vs. emotional reduction approaches to chronic pain due to primary headache. Thus, to enhance the positive effects of all these treatments, future research might consider pairing treatments aimed at relaxation together with treatments aimed at emotional activation, management, and control, not so much to compare the outcomes on emotional management, but to strengthen the effectiveness of the combined treatments.

The present study has some limitations. First, the literature reviewed is limited to the last 23 years (2000–2023), so it cannot discuss the literature beyond that period. Second, this review presents some restrictions placed on the language (English) and access to full text, and it allowed that some articles should be excluded from the retrieved ones. Third, we did not consider psychotherapeutic treatments as alternative interventions, for which special attention would be needed in a future review. Finally, the studies are methodologically heterogeneous, so we did not assess their quality.

## 5. Conclusions

The results summarized in the present review contribute to the enrichment of knowledge concerning the effects of some alternative treatments on the management and processing of emotions in patients with primary headache. The retrieved articles focused on RT, WED, different forms of meditation, written emotional disclosure, AAET, and CFT. The results highlight the effects of each treatment alone or in combination with others on headache symptoms, negative emotions, pain tolerance, emotion control, and mood. But, beyond the specific effects of each treatment, these non-pharmacological interventions represent tools that can be used by clinicians on patients suffering from chronic pain due to primary headache in accordance with medical ones. These interventions are techniques that patients can practice at home without special equipment or financial commitment, with benefits in terms of health care costs compared to pharmacological treatments, as well as increasing the patient’s self-efficacy and control in pain management, as well as emotion management. Therefore, we hope that future research will take into account the potential and limitations previously described about each alternative treatment considered, with special attention being paid to their methodological aspects, and will continue the in-depth investigation of these treatments in order to increasingly implement and disseminate them in clinical practice.

## Figures and Tables

**Figure 1 healthcare-12-01302-f001:**
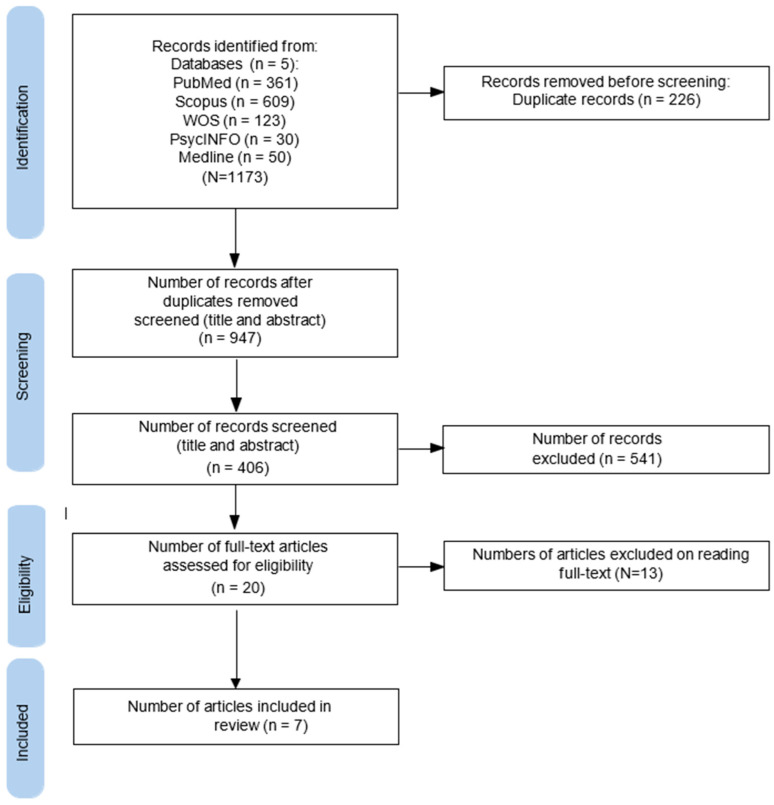
Flowchart of study selection.

**Table 1 healthcare-12-01302-t001:** General characteristics of included studies.

Author (Reference), Year (Country)	Study Design and AIM	Participants: Sample Size, (Mean Age, SD), Females (%)	Headache Type	Treatment	Procedure/Tools	Emotions or Affects Measure/Collection	Primary Outcome	Effects on Emotional Expression or Management
Wachholtz et al., 2019 (USA) [29]	Longitudinal randomized controlled trial study Aimed to compare the effects of three active management forms of meditation vs. a cognitive distraction meditation on headache-related pain and emotions	80 college students (19 years, SD = 1.06), 90% female	Migraine headache. Eligibility criteria: at least two episodes in the previous month. Participants were selected by the self-screener ID Migraine Screener	Active management group (spiritual meditation; internally focused secular meditation; progressive muscle relaxation) vs. cognitive distraction group (external focused distraction phrases)	Participants were asked to practice the technique for 20 min per day, for 30 days. They completed daily diaries (headache logs and emotion evaluations)	A self-report seven-point scale to rate happiness, sadness, calmness, and anger	Self-reported severity of migraine headache and felt emotions across the 30 days of the diary	The active management group showed significant decreases in anger (*p* = 0.005) and migraine pain (*p* = 0.002). The bulk of the change for the active management group occurred in the final 10 days, after 20 days of practice of the technique (*p* < 0.05). The cognitive distraction group’s reported anger and pain did not change over 30 days.
Slavin-Spenny et al., 2013 (USA) [30]	A randomized trial. Aimed to compare the effects of a brief group-based anger awareness and expression training course to relaxation training on headache management and negative emotions	147 college students (21.1, SD = 6.0), 87.8% female	Headache (mixed with respect to headache type). The diagnoses were self-reported, and only 26.7% of participants declared they had been diagnosed with migraine by a physician.	Anger awareness and expression training (AAET) vs. relaxation training (RT) vs. a waist-list control	Participants were randomly assigned to the three conditions and attended in three treatment sessions for the two experimental conditions.	The 20-item Toronto Alexithymia Scale; the 30-item Rathus Assertiveness Schedule; and the 4-item emotional approach coping scale	Headache frequency; headache severity and duration; headache disability; and psychological symptoms	The outcome measures of the two interventions are very similar. For manipulation check affect reactions, the interventions differed in the change in negative affect, t(76) = 3.01, *p* = 0.004; the RT condition reduced negative affect (M = −1.38, SD = 0.96) more than the AAET condition (M = −0.58, SD = 1.35), although negative affect decreased significantly in both conditions (*p* < 0.001 and *p* = 0.01, respectively). Emotional expression did not change over time for any of the conditions.
Tonelli and Wachholts, 2014 (USA) [31]	Clinical intervention-based study (single-group repeated-measures, pre/post design). Aimed to explore the efficacy of the Buddhist Loving Kindness meditation practice in reducing both migraine-headache symptoms and emotional tension (negative affective state).	27 (general population) (45.5, SD = 11.10), 68% female	Migraine headache. Eligibility criteria: participants have ten migraine episodes per month; migraine diagnosis is confirmed by a primary care provider. Participants completed the three-item ID Migraine Screener	Meditation intervention (Buddhist Loving Kindness meditation practice)	Participants completed a pre-intervention survey, and after each meditation session they repeated the survey, rating their level of migraine-related pain and emotion tension. Group-based 20 min guided meditation session.	The Numeric Rating Scale (NRS-11) in its original form to assess pain was used, and it was altered to assess the emotional tension.	Pain and emotional tension were assessed using The Numeric Rating Scale (NRS-11).	Meditation reduces migraine symptoms and saw a decrease in reported levels of emotional tension. After meditation, both reported pain levels (mean 2.62, SD 1.713) and reported emotional tension levels (mean 2.27, SD 2.187) decreased. Data before and after meditation indicated a 32.7% decrease in pain and a 42.7% decrease in emotional tension.
Wachholtz and Pargament, 2008 (USA) [32]	A randomized trial. Aimed to compare spiritual meditation to secular meditation and relaxation training in enhancing pain tolerance and reducing migraine–headache-related symptoms.	83 college students (19.1, SD = 1.10), 90.4% female	Vascular headache (migraine; mixed migraine + tension headache). Participants completed the three-item ID Migraine Screener	Spiritual meditation (SM), internally focused secular meditation (IFSM), external secular meditation (ESM), focused secular meditation (FSM), muscle relaxation (MR)	Participants were randomly assigned into four groups. They practiced 20 min a day for one month the type of meditation or relaxation training their group was assigned.	The Positive and Negative Affect Scale (PANAS Scale)	Pain tolerance, headache frequency, mental and spiritual variables, and psychological measures (affect, anxiety, depression)	Spiritual meditation contributes to a decrease in the frequency of migraine, anxiety, and negative affect, as well as a greater increase in pain tolerance, headache-related self-efficacy, daily spiritual experiences, and existential well-being. For negative affect, the SM group reported a significantly greater drop in negative affect over the course of the study than the ISM (*p* < 0.001; g2 = 0.25) or ESM (*p* < 0.01; g2 = 0.15) groups. For positive affect, three of the four groups (SM, ISM, and MR) experienced some modest improvement in their positive affect; this was not significant (F (3.79) = 0.26, *p* = NS, g2 = 0.01).
D’Souza et al., 2008 (USA) [33]	A randomized trial. Aimed to compare the effects of relaxation training (RT) and written emotional disclosure (WED) with respect to mood, headache frequency, severity, disability, and physical symptoms.	141 college students, 85.9% female 51 with headache tension (20.27, SD = 2.30); 90 with migraine headache (21.44, SD = 5.47)	Tension headache and migraine headache (at least two episodes per week, moderate or severe intensity). Participants self-reported headache type and frequency	Relaxation training (RT), written emotional disclosure (WED)	Four repeats of 20 min sessions over two consecutive weeks for each condition (relaxation training, emotional disclosure, neutral writing, and control conditions). Before and after each session, participants rated their mood. They also returned to the laboratory to complete their health status at 1 month and 3 month follow-ups.	The Positive and Negative Affect Scale (PANAS Scale)	Headache frequency; headache severity; headache disability; and physical symptoms	RT was effective for tension headaches, but WED had no effect on health status for either tension or migraine headaches. For mood, RT led to an immediate increase in calmness (M = 1.74, SD = 1.08) more than the control group (M = 0.09, SD = 0.96), t(57) = 6.15, *p* < 0. 001, pη2 = 0.40, and more than the WED group (M = −0.71, SD = 1.31), t(56) = 7.47, *p* < 0.001, pη2 = 0.50. Whereas, WED led to an immediate increase in negative mood (M = 0.51, SD = 0.88) compared with both the controls (M = −0.16, SD = 0.39), t(59) = 4.00, *p* < 0.001, pη2 = 0.21 and the RT group (M = −0.64, SD = 0.49), t(56) = 6.23, *p* < 0.001, pη2 = 0.41.
Kraft et al., 2008 (USA) [34]	A randomized trial. Aimed to compare the written emotional disclosure with control writing and relaxation training, and to test the moderate effects of emotional skills and headache management self-efficacy.	90 college students (21.4, SD = missing), 88.8% female	Migraine headache (at least one episode per month). Participants had a diagnostic interview to confirm migraines.	Relaxation training (RT), written emotional disclosure (WED), a control writing condition	Four repeats of 20 min lab sessions over two weeks. Follow-up assessments at 1 and 3 months.	Eight-item emotional approach coping scale (EAC).	Headache frequency, pain severity, functional and emotional disability, and negative and positive affect for the past month	Greater emotional approach coping predicted improvement following WED compared to RT and the control, whereas low headache management self-efficacy predicted an improvement following both WED and RT, compared to the control.
Barchakh et al., 2020 (Iran) [35]	Quasi-experimental. Convenience sampling method. Aimed to investigate the effectiveness of compassion-focused therapy for improving emotional control and reducing the severity of pain in patients with migraines.	30 patients (34.43, SD = 11.07), 80% female	Migraine headache. Participants were evaluated using the Migraine Disability Assessment (MIDA) to verify their migraines.	Compassion-focused therapy (CFT)	Participants were randomly assigned to experimental and control groups. Eight 90 min sessions of CFT (experimental group). Post-test after finishing the training sessions.	Emotion Control Questionnaire (ECQ).	Emotional control and severity of pain.	The CFT training improves emotional control (F = 21.81; *p* < 0.01) and reduces the severity of the pain in patients with migraines (F = 17.21; *p* < 0.01). The CFT should be considered effective for improving the emotional control of patients and reducing pain.

**Table 2 healthcare-12-01302-t002:** A summary of major limitations.

Major Limitations	NO. of Studies	Study ID
Small sample.	6	29, 30, 31, 32, 33, 34
The study population consisted mainly of young adults, university students, and undergraduates (not a clinical sample).	6	29, 30, 31, 32, 33, 34
Sample made up exclusively of migraine patients.	1	35
Sample consisting mostly of women.	3	29, 32, 34
The headache diagnoses reported by the patients were not verified by a pain specialist.	3	29, 30, 32
All outcome measures were retrospective self-reports.	1	30
The follow-up period was too short to assess the effects of treatment in the long term.	1	30
The study only relied on subjective measures and not objective measures to detect levels of pain and emotional tension.	1	31
The study maintained minimal contact between the participants and the experimenter.	1	32
No follow-up.	1	32

## Data Availability

Not applicable.

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
