# Peer review of "Alternative Treatments for Emotional Experiencing and Processing in People with Migraine or Tension-Type Headache: A Scoping Review"

_healthcare, 2024, doi:10.3390/healthcare12131302_

Round 1
Reviewer 1 Report
Comments and Suggestions for Authors
An interesting article. I only have a few comments:
Row 162: you miss the bracket [36].
Row 168.170: your paper will gain, if you give a better description of RT and WED, which have a central place.
Row 237-42: In the parenthesis you use e.g., which means for example. Should it be i.d.? which means id est, that is.
Row 304-5: It is unclear, what do you mean? Decreases or increases. Re-write the sentence.
Author Response
Dear reviewer,
Thank you very much for your comments and suggestions, which are valuable in improving the quality of our manuscript.
Below are the answers to all your questions.
In the manuscript, all corrections are underlined in yellow.
Best regards,
Alessandra Marelli
Reviewer 1
An interesting article. I only have a few comments:
Row 162: you miss the bracket [36].
Reply: we added the bracket (Row 158)
Row 168.170: your paper will gain if you give a better description of RT and WED, which have a central place.
Reply: we added a better description of RT, WED, and other alternative treatments considered in the articles selected (Row: 168-202).
Row 237-42: In the parenthesis, you use e.g., which means for example. Should it be i.d.? which means id est, that is.
Reply: we have deleted e.g. (Row: 268)
Row 304-5: It is unclear, what do you mean? Decreases or increases. Re-write the sentence.
Reply: we reviewed the entire discussion section and deleted the sentence.

Reviewer 2 Report
Comments and Suggestions for Authors
The present review aims to summarize the effect of non-pharmacological treatments (e.g. relaxation training, meditation, written intervention) on emotional expression, processing, control, or management in patients with migraine and tension-type headaches. The authors performed search in 5 online databases (PubMed, Scopus, Web of Science, PsycINFO, and Medline) to identify studies published between 2000 and may of 2023. They included 7 articles after screening 1.173 records. A total of 610 patients with a diagnosis of migraine or tension-type headache, and average age 19–45.5 years (68–90.4% females) were recruited in the 16 selected studies. Overall, the results show that non-pharmacological alternative approaches to headache treatment contribute to the management, reduction, or control of negative emotions and at the same time have a positive impact on pain perception and headache symptoms.
The authors describe the selected papers and discuss them adequately according to the previous literature. In particular, I found the topic of their article very interesting and necessary to verify whether these non-pharmacological strategies are effective or not. I have only one major and two minor suggestions:
- Major: The authors need to update the search by at least the end of 2023. Ideally, if possible, they can update until April 2024.
- Minor:
Why not do a systematic review instead of a narrative review?
Overall, the authors need to break up the paragraphs into smaller ones. The actual paragraphs are too large throughout the manuscript.
Author Response
Dear reviewer,
Thank you very much for your comments and suggestions, which are valuable in improving the quality of our manuscript.
Below are the answers to all your questions.
In the manuscript, all corrections are underlined in yellow.
Best regards,
Alessandra Marelli
Reviewer 2
The present review aims to summarize the effect of non-pharmacological treatments (e.g. relaxation training, meditation, written intervention) on emotional expression, processing, control, or management in patients with migraine and tension-type headaches. The authors performed search in 5 online databases (PubMed, Scopus, Web of Science, PsycINFO, and Medline) to identify studies published between 2000 and may of 2023. They included 7 articles after screening 1.173 records. A total of 610 patients with a diagnosis of migraine or tension-type headache, and average age 19–45.5 years (68–90.4% females) were recruited in the 16 selected studies. Overall, the results show that non-pharmacological alternative approaches to headache treatment contribute to the management, reduction, or control of negative emotions and at the same time have a positive impact on pain perception and headache symptoms.
The authors describe the selected papers and discuss them adequately according to the previous literature. In particular, I found the topic of their article very interesting and necessary to verify whether these non-pharmacological strategies are effective or not. I have only one major and two minor suggestions:
- Major: The authors need to update the search by at least the end of 2023. Ideally, if possible, they can update until April 2024.
Reply: We update the search in the selected databases from May 2023 to May 2024. Only one additional study was found*, but it was not included in the scoping review because it focuses on psychotherapy intervention, which was not considered within the inclusion criteria of selection studies. As suggested by another reviewer, we specified that our review did not focus on psychotherapeutic intervention. So we didn't update the time interval of study selection.
*Shahverdi, Z. A., Dehghani, M., Ashouri, A., Manouchehri, M., & Mohebi, N. (2024). Effectiveness of intensive short-term dynamic psychotherapy for Tension-Type Headache (TTH): A randomized controlled trial of effects on emotion regulation, anger, anxiety, and TTH symptom severity. Acta Psychologica, 244, 104176.
- Minor: Why not do a systematic review instead of a narrative review?
Reply: In accordance with Munn et al. (2018)* and Smirth&Duncan (2022)**, we decided to conduct a scoping review because we started with a broad research question and we would like to identify and analyze knowledge gaps or to identify key characteristics or factors related to a concept. In addition, a scoping review methodology is appropriate to research the types of evidence available and corresponding research methods in a particular field, to clarify key concepts to identify related factors, and to analyze knowledge gaps. Our scoping review aimed to focus attention on the procedures and methods adopted in the selected papers, and also to give useful directions for future research in this area.
*Munn, Z., Peters, M. D., Stern, C., Tufanaru, C., McArthur, A., & Aromataris, E. (2018). Systematic review or scoping review? Guidance for authors when choosing between a systematic or scoping review approach. BMC medical research methodology, 18, 1-7.
**Smith, S. A., & Duncan, A. A. (2022, December). Systematic and scoping reviews: A comparison and overview. In Seminars in Vascular Surgery (Vol. 35, No. 4, pp. 464-469). WB Saunders.
Overall, the authors need to break up the paragraphs into smaller ones. The actual paragraphs are too large throughout the manuscript.
Reply: We reviewed and summarized the introduction, results, and discussion sections.

Reviewer 3 Report
Comments and Suggestions for Authors
Thank you for the opportunity to review your interesting paper.
My comments on the manuscript (healthcare-2958580) are as follows.
# Overall opinion
As a scoping review on an interesting topic, it is worth considering for publication.
However, the following issues need to be addressed before publication.
# Recommendations for revision
1. very long introduction in one paragraph, which reduces readability. It should be split into about 3 paragraphs and be more explicit about what you want to cover in each paragraph. The "context of this research" should be clearer than it is.
2. as a separate section, the nature of the intervention and its defined benefits should be more specifically described, and the comparator to which each intervention was compared in each of the few trials should be reported in detail.
3. some quantitative analysis of outcomes is essential; currently the manuscript is too qualitative and it is difficult to determine the effect size of the specific CAMs discussed in this manuscript and whether they are worthy of further study.
4. The design of the individual studies included in this manuscript should also be specifically addressed in a separate section.
5. as a complement to the above, conclusions regarding the "quantitative metrics" obtained from this manuscript should be concisely presented at the end of the RESULTS section, along with qualitative conclusions that include the authors' views.
Author Response
Dear reviewer,
Thank you very much for your comments and suggestions, which are valuable in improving the quality of our manuscript.
Below are the answers to all your questions.
In the manuscript, all corrections are underlined in yellow.
Best regards,
Alessandra Marelli
Reviewer 3
Thank you for the opportunity to review your interesting paper.
My comments on the manuscript (healthcare-2958580) are as follows.
# Overall opinion
As a scoping review on an interesting topic, it is worth considering for publication.
However, the following issues need to be addressed before publication.
# Recommendations for revision
- very long introduction in one paragraph, which reduces readability. It should be split into about 3 paragraphs and be more explicit about what you want to cover in each paragraph. The "context of this research" should be clearer than it is.
Reply: We apologize for sending the introduction without any separation into paragraphs. We have divided the introduction into different parts.
In accordance with other reviewers' comments, we revised the entire introduction.
- as a separate section, the nature of the intervention and its defined benefits should be more specifically described, and the comparator to which each intervention was compared in each of the few trials should be reported in detail.
Reply: we rewrote the results, describing the interventions and their benefits in comparison with each treatment.
- some quantitative analysis of outcomes is essential; currently the manuscript is too qualitative and it is difficult to determine the effect size of the specific CAMs discussed in this manuscript and whether they are worthy of further study.
Reply: As a scoping review, we did not report the effects of the treatments but a qualitative commentary. However, we included some values of the effects found by the various treatments in Table 1.
- The design of the individual studies included in this manuscript should also be specifically addressed in a separate section.
Reply: we rewrote the results, adding more details to the research design and methodological procedure.
- as a complement to the above, conclusions regarding the "quantitative metrics" obtained from this manuscript should be concisely presented at the end of the RESULTS section, along with qualitative conclusions that include the authors' views.
Reply: we reviewed the results section, specifying the results of the studies more precisely. We included the effects obtained in the various treatments in Table 1 too.

Reviewer 4 Report
Comments and Suggestions for Authors
The paper by Marelli et al. deals with the review of some non-pharmacological interventional techniques to be applied in primary headaches. The paper is interesting and the subject is quite promising, yet I have some doubts I would like to ask to authors:
1) The title sounds too catchy and wide compared with the narrow type of intervention considered: the paper reviewed only those non-pharmacological, non-psychotherapeutic interventions whose mechanism of action relies on modifying emotions. I think that it should be turned down a little.
2) As well in the text, authors should refer to something more limited than “non-pharmacological therapies” explain that they focused on a precise share of all non-pharmacological interventions
3) In the introduction or discussion, among non-pharmacological options psychodynamic psychotherapy have not be mentioned while there is a small but consistent number of evidence of its effect. In particular, it also acts on emotions. I think that this could be worth especially when the counter-intuitive effect of WED is discussed.
4) Discussion (and conclusions too) is, in my opinion, too long. Could it be shortened?
5) In limitations authors state (lines 500-503) that the large and unselected recruitment of samples in studies affects generalizability. In my opinion, this is the opposite. It reduced chances to see a result but increases generalizability. In case it reduced the chances to apply these results to patients recruited in headache centers. The following sentence (line 504-505) in this sense sounds much more in line with my idea. As well, the high percentage of women is not surprising in headache field. I think that small number of patients, non-standard timing and duration of interventions, inclusion criteria mixing low-, high- frequency and chronic migraine could be more impactful as limitations
The ref 8 primarily refers to the different effect of pharmacological treatments based on gender disparity while it is cited for enlisting non pharmacological therapies. Although very interesting as a point, I think there is some more focused citations for this statement.
Line 162 the citation 36 has no brackets and line 163 contains a repetition of “and”
Comments on the Quality of English Language
About style, I would recommend to reduce the number of brackets sections. They are generally redundant and slow down the pace of the reading. Often the term or the content of the brackets section fits better.
Author Response
Dear reviewer,
Thank you very much for your comments and suggestions, which are valuable in improving the quality of our manuscript.
Below are the answers to all your questions.
In the manuscript, all corrections are underlined in yellow.
Best regards,
Alessandra Marelli
Reviewer 4
The paper by Marelli et al. deals with the review of some non-pharmacological interventional techniques to be applied in primary headaches. The paper is interesting and the subject is quite promising, yet I have some doubts I would like to ask to authors:
1) The title sounds too catchy and wide compared with the narrow type of intervention considered: the paper reviewed only those non-pharmacological, non-psychotherapeutic interventions whose mechanism of action relies on modifying emotions. I think that it should be turned down a little.
Reply: We modified the title “Alternative treatments for emotional processing in people with migraine or tension-type headache: a scoping review”, and we added additional information in the introduction section and the study selection section about the exclusion of non-psychotherapeutic interventions.
2) As well in the text, authors should refer to something more limited than “non-pharmacological therapies” explain that they focused on a precise share of all non-pharmacological interventions
Reply: We made it clear that we did not focus on psychotherapeutic treatments.
3) In the introduction or discussion, among non-pharmacological options psychodynamic psychotherapy have not be mentioned while there is a small but consistent number of evidence of its effect. In particular, it also acts on emotions. I think that this could be worth especially when the counter-intuitive effect of WED is discussed.
Reply: We mentioned psychodynamic psychotherapy in the introduction section, and we rewrote the discussion. To try to reduce the discussion we have greatly reduced the part about WED, so we did not add other comments about psychodynamic psychotherapy.
4) Discussion (and conclusions too) is, in my opinion, too long. Could it be shortened?
Reply: We rewrote the discussion and conclusion sections.
5) In limitations authors state (lines 500-503) that the large and unselected recruitment of samples in studies affects generalizability. In my opinion, this is the opposite. It reduced chances to see a result but increases generalizability. In case it reduced the chances to apply these results to patients recruited in headache centers. The following sentence (line 504-505) in this sense sounds much more in line with my idea. As well, the high percentage of women is not surprising in headache field. I think that small number of patients, non-standard timing and duration of interventions, inclusion criteria mixing low-, high- frequency and chronic migraine could be more impactful as limitations.
Reply: Sorry, but we did not find where we referred to a large sample. In any case, we completely rewrote the discussion and we added Table 2 about the studies’ limitations.
The ref 8 primarily refers to the different effect of pharmacological treatments based on gender disparity while it is cited for enlisting non pharmacological therapies. Although very interesting as a point, I think there is some more focused citations for this statement.
Reply: We revised the entire introduction and removed that reference.
Line 162 the citation 36 has no brackets and line 163 contains a repetition of “and”
Reply: we added the bracket to the citation 36 (Row 158); we removed the repetition “and” (Row 158-159)
Comments on the Quality of English Language
About style, I would recommend to reduce the number of brackets sections. They are generally redundant and slow down the pace of the reading. Often the term or the content of the brackets section fits better.
Reply: We reduced the brackets sections.

Round 2
Reviewer 2 Report
Comments and Suggestions for Authors
I suggest publication